Taxonomic identification, genomic analysis, and optimized chromium(VI) bioreduction by Microbacterium triticisoli sp. nov. M28T

Qing Yu 1
Tian Jiewei 1 2
Ma Zhenhua 1
Tang Miao 1
Long Xiufeng 1 longxiufeng@gxust.edu.cn
1 Guangxi University of Science and Technology , Liuzhou , China
2 Jinan Fruit Research Institute , Jinan , China
Wang Liang
Electronic publication date: 2025 Oct 23
Publication date: 2025
Volume: 13
Electronic Location ID: e20192
Received 2025 Jul 1; Accepted 2025 Sep 15
Copyright: © 2025 Qing et al.
Copyright year: 2025
Copyright holder: Qing et al.
License: This is an open access article distributed under the terms of the Creative Commons Attribution License, which permits unrestricted use, distribution, reproduction and adaptation in any medium and for any purpose provided that it is properly attributed. For attribution, the original author(s), title, publication source (PeerJ) and either DOI or URL of the article must be cited.
License URL: https://creativecommons.org/licenses/by/4.0/

Keywords: Microbacterium triticisoli sp. nov., Polyphasic taxonomy, Genome analysis, Cr(VI) reduction

Funding: Guangxi Natural Science Foundation 2024GXNSFBA010351 and 2025GXNSFHA069093 Guangxi University of Science and Technology Doctoral Fund 20Z18 This work was funded by the Guangxi Natural Science Foundation (No. 2024GXNSFBA010351 and 2025GXNSFHA069093), and Guangxi University of Science and Technology Doctoral Fund (No. 20Z18). The funders had no role in study design, data collection and analysis, decision to publish, or preparation of the manuscript.

==============================
Strain M28T was isolated from subsoil obtained from decaying wheat straw. Cells were Gram-positive, non-motile, short rod-shaped and formed yellowish colonies on lysogeny broth (LB) agar. The strain was able to grow at 0–8% (w/v) NaCl , 15–40 °C and pH 5.5–10.0. Phylogenetic analysis based on 16S rRNA gene sequences, core genes and whole-genome indicated that strain M28T belonged to the genus Microbacterium but was distinct from all known strains in this genus. Based on phenotypic, genotypic, chemical taxono mic and phylogenetic analyses, strain M28T is a representative of a new species of Microbacterium, which is proposed to be named Microbacterium triticisoli sp. nov., the type strain is M28T (=CCTCC AA 2022021T =JCM 35796T). Genomic analysis revealed multiple metal resistance systems, antibiotic resistance determinants and oxidative stress defense genes, explaining its exceptional environmental adaptability. Notably, the strain reduced 99% of 50 mg/L Cr(VI) within 24 h under optimized conditions (37 °C, pH 7.0, 2.5 g/L sucrose) and tolerated Cr(VI) concentrations up to 125 mg/L. This study identifies M. triticisoli as a promising agent for chromium bioremediation, providing a foundation for engineering microbial solutions to heavy metal pollution.

Introduction

In recent years, the accelerated development of agricultural, industrial, and urban sectors has led to a marked increase in chromium contamination within wastewater systems. This issue is particularly pronounced in tannery operations, electroplating facilities (Liang et al., 2021), and pigment production plants, which constitute primary contributors to chromium pollution (Sharma et al., 2022). Such human activities have caused significant environmental degradation in affected regions and widespread societal concerns regarding chromium contamination (Xiao et al., 2024). In aqueous environments, hexavalent chromium predominantly exists as Cr2O72−, CrO42−, H2CrO4, and HCrO4−, with its speciation dynamics governed by redox potential (Eh) (Zhou et al., 2024), solution pH, total chromium concentration, and the presence of oxidizing/reducing agents (Chen et al., 2024). In contrast, trivalent chromium (Cr(III)) exhibits significantly lower toxicity, primarily occurring as Cr3+ and CrO2−. These species demonstrate strong adsorption affinities toward soil colloids and tend to form insoluble amorphous hydroxide complexes under natural conditions (Richard & Bourg, 1991). Although Cr(III) functions as an essential trace element in mammalian physiology by enhancing insulin-receptor interactions for glucose homeostasis (Hossini et al., 2022; Vincent, 2024), prolonged exposure to sources originating from human activities may still induce dermatological irritation and potential carcinogenic effects (Georgaki & Charalambous, 2023; Shin et al., 2023). Given these health concerns, the International Agency for Research on Cancer (IARC) reclassified Cr(VI) as a Group 1 human carcinogen in 2016, notably with its toxicity potency 100–1,000 times greater than Cr(III) (Jia et al., 2018). Subsequent investigations, including a 2019 U.S. epidemiological study, have established robust correlations between elevated groundwater Cr(VI) levels and anthropogenic pollution sources, underscoring the persistent environmental risks associated with this heavy metal (Xia et al., 2019). Given these concerns, developing ecotechnological strategies for Cr(VI) remediation has become imperative. Current evidence indicates that reducing toxic Cr(VI) to environmentally benign Cr(III) through valence state conversion represents a highly effective remediation strategy (Tumolo et al., 2020).

Numerous studies have demonstrated that bioreduction serves as the primary mechanism for environmental Cr(VI) removal (Chen et al., 2021a). The bioremediation of Cr(VI), particularly through functional microorganisms, has emerged as an optimal strategy for chromium elimination owing to its cost-effectiveness, high efficiency, and lack of secondary pollution (Karimi-Maleh et al., 2021). Diverse Cr(VI)-reducing microorganisms have been identified in coastal ecosystems, including Enteromorpha prolifera (Kalsoom et al., 2023), Bacillus sp. (Chen et al., 2021b), Staphylococcus aureus (Wang et al., 2010), Aspergillus sp. (Gu et al., 2015), Desulfovibrio sp. (Goulhen et al., 2006), Exiguobacterium sp. (Huang et al., 2023), Rhodococcus sp. (Sun et al., 2011), Enterobacter sp. (Rahman & Singh, 2014), and Leucobacter sp. (Tahri Joutey et al., 2016). These microbial strains enzymatically convert highly toxic Cr(VI) into low-toxicity, readily precipitable Cr(III) through enzymatic catalysis or extracellular electron transfer mechanisms, thereby significantly reducing heavy metal ecotoxicity. Notably, specific strains (e.g., Bacillus and Rhodococcus) demonstrate remarkable resistance to complex polymetallic contamination, enabling synergistic heavy metal remediation through metabolic interactions. Consequently, the screening of microbial species exhibiting both Cr(VI) tolerance and rapid reduction capacity constitutes a critical approach for mitigating chromium pollution impacts.

The genus Microbacterium (Family Microbacteriaceae, Order Actinomycetales, Class Actinobacteria) was first taxonomically delineated by Orla-Jensen (1919). Subsequent emendations by Collins, Jones & Kroppenstedt (1983) refined the genus circumscription through DNA-DNA hybridization and chemotaxonomic characteristics (including cell wall composition, menaquinone profiles, and fatty acid profiles), which excluded atypical strains and established more precise defining characteristics. The most recent taxonomic refinement was achieved through polyphasic characterization by Takeuchi & Hatano (1998), who resolved phylogenetic inconsistencies by synonymizing Microbacteria sp. within the Microbacterium clade. According to the List of Prokaryotic Names with Standing in Nomenclature (LPSN), this genus currently comprises 129 validly published species. Continuous taxonomic exploration since 2020 has expanded its ecological range, with novel species isolated from soil (Liu et al., 2022), water (Lee et al., 2024), dairy (Bellassi et al., 2021), inter-root (Deng et al., 2024), faeces (Dong et al., 2020) and extreme environments (Qiu et al., 2024). Isolations from such diverse sources demonstrated how environmental heterogeneity drives niche-specific genomic adaptations across this genus. Comparative genomics reveals horizontal gene transfer events enabling phenotypic plasticity across ecological gradients (Yuan et al., 2024). Microbacterium sp. exhibit remarkable bioremediation potential through: As(III) oxidation (Sher et al., 2022), Cr(VI) reductase activity (Mishra et al., 2021), Cd(II) efflux systems (Ma, Wang & Zhang, 2023; Long et al., 2021), PAH dioxygenase pathways (Logeshwaran et al., 2022), Organophosphate hydrolases (Logeshwaran et al., 2020), Alkane hydroxylase complexes (Hazaimeh et al., 2024), Biogeochemical cycling, Iron-sulfur redox coupling, Phosphorus solubilization and so on. Members of the genus Microbacterium are known for their metabolic diversity, especially in heavy metal tolerance, organic pollutant degradation and biogeochemical cycling, and many of these strains have been shown to degrade arsenic, chromium, cadmium, polycyclic aromatic hydrocarbons (PAHs), pesticides and petroleum hydrocarbons.

In this study, chromium-reducing microbial strains were isolated and screened from contaminated soils, subsequently characterizing its taxonomy through polyphasic analysis and whole-genome sequencing (Illumina MiSeq platform), interrogating key functional genes, and systematically evaluating its hexavalent chromium reduction capacity.

Materials and Methods

Isolation and screening of Cr(VI)-resistant strains

The chromium-tolerant strain was isolated from soils collected beneath decomposing decayed wheat straw residues in Qingyang, Gansu Province, China (35°45′N, 107°36′E). A 3 g soil was aseptically transferred into 100 mL of sterile chromium-supplemented minimal medium (composition per liter: 0.5 g K2HPO4, 0.5 g KH2PO4, 0.5 g NaCl, 0.5 g MgSO4·7H2O, 2 g (NH4)2SO4, 0.2829 g K2Cr2O7; pH 7.0), followed by 24 h incubation at 37 °C with orbital shaking (150 rpm). After primary enrichment, the supernatant was subjected to ten-fold serial dilution using sterile phosphate-buffered saline (pH 7.4). Aliquots (100 μL) from the 10−6, 10−7, and 10−8 dilutions were aseptically plated onto selective lysogeny broth (LB) agar medium containing Cr(VI) (per liter: 10 g tryptone, 5 g yeast extract, 1 g NaCl, 0.2829 g K2Cr2O7, 15 g agar; pH 7.0). Plates were incubated under aerobic conditions at 37 °C for 72 h. Distinct colonies exhibiting morphological diversity were subcultured through on fresh Cr(VI)-amended LB agar. Following three successive purification cycles, multiple Cr(VI)-resistant isolates with stable colonial characteristics were obtained for subsequent characterization.

Phylogenetic characterization was initiated by amplifying the nearly full-length 16S rRNA gene sequence using universal prokaryotic primers 27F (TACGGYTACCTTGTTACGACTT) and 1492R (AGAGTTTGATCMTGGCTCAG) under standardized PCR conditions. PCR amplification was performed in a 50 μL reaction volume consisting of: 25 μL commercial PCR premix, 1 μL each of forward and reverse primer (10 μM), 1 μL bacterial culture, and 22 μL sterile distilled water. The amplification protocol comprised: an initial denaturation at 95 °C for 5 min; followed by 30 cycles of denaturation (95 °C for 30 s), annealing (56 °C for 30 s), and extension (72 °C for 90 s); with a final extension at 72 °C for 5 min. The PCR amplification products were sent to Sangon Biotech (Shanghai) Co., Ltd. for sequencing using the Sanger sequencing method, and then uploaded to NCBI and EzBiocloud (https://www.ezbiocloud.net/) for alignment. The sequence similarity of strain M28T showed that it was a suspected new species, so strain M28T was selected for the subsequent polyphasic taxonomic identification and Cr(VI) reduction study.

The methods for analyzing the phenotypic characteristics, physiological and biochemical characteristics, and chemical components of strain M28T were provided in the supplemented materials.

Whole-genome sequencing and the phylogenetic analysis

Genomic DNA of strain M28T was extracted using the Extraction Kit (magnetic bead method, product no. T07-100; Shanghai Meiji Biological, Shanghai, China). A 50 mL culture of bacteria in the logarithmic growth phase (approximately 0.8 g wet cell weight) was centrifuged at 12,000 rpm to harvest the cells. Subsequent steps followed the manufacturer’s protocol: lysis was performed with 1.8 mL of buffer TL1 for 30 min at 70 °C, followed by magnetic bead-based purification and elution, yielding over 40 μg of high-purity DNA. Whole-genome sequencing and assembly were performed at Shanghai Majorbio Bio-pharm Technology Co., Ltd. through a combination of PacBio RS II single-molecule real-time (SMRT) sequencing and Illumina NovaSeq 6000 platforms (San Diego, USA). Illumina raw data were filtered using fastp v0.23.0 (trimming the first and last 10 bp of low-quality bases, removing reads with Q < 20 and N > 5%), retaining 150 bp paired-end clean data of 3.2 Gb (30% of the original data volume). Nanopore data and Illumina clean data were mixed and assembled using Unicycler v0.4.8 to construct a chromosome-level continuous sequence. Sequencing data were assessed for genome size, and contamination. Genome assembly utilized: Fastp (https://github.com/OpenGene/fastp), SOAPdenovo2 (https://anaconda.org/bioconda/soapdenovo2) (Luo et al., 2012), Unicycler v0.4.8 (Wick et al., 2017), Pilon v1.22. Chromosomal gene prediction was conducted using: Glimmer (http://ccb.jhu.edu/software/glimmer/index.shtml) (Delcher et al., 2007), tRNAscan-SE v2.0 (http://trna.ucsc.edu/software/) (Benson, 1999), Barrnap (https://github.com/tseemann/barrnap) (Liu et al., 2019). The circular chromosome of strain M28T was visualized using CGView (Alanjary, Steinke & Ziemert, 2019). Functional annotation of the genome was conducted against the Gene Ontology (GO) (Gene Ontology Consortium, 2004), Non-Redundant Protein (NR) (Qi, Lee & Hayward, 2005), Swiss-Prot (Boeckmann et al., 2003), Pfam (Finn et al., 2014), Clusters of Orthologous Groups (COG) (Koonin, 2002), and Kyoto Encyclopedia of Genes and Genomes (KEGG) (Kanehisa & Goto, 2000) databases.

The 16S rRNA gene sequence of strain M28T was deposited in the EzBioCloud and NCBI database for phylogenetic identification (The GenBank accession number for the 16S rRNA gene is ON923955). Phylogenetic trees were reconstructed using MEGA 7 software with maximum-likelihood (ML), neighbor-joining (NJ), and minimum-evolution (ME) algorithms. To further resolve taxonomic placement, core genome analysis was performed through: (1) MiGA-based and autoMLST-driven extraction of core genes/proteins; (2) ML phylogenetic reconstruction using MEGA 7. Whole genome sequence data were submitted to the Type Strain Genome Server (TYGS; https://tygs.dsmz.de) for genome-based taxonomic analysis following established pipelines (Meier-Kolthoff et al., 2022). Reference genomes were acquired from NCBI RefSeq database. Genomic relatedness was assessed through: (i) digital DNA-DNA hybridization (dDDH) using the Genome-to-Genome Distance Calculator (GGDC 3.0) with BLAST+ alignment; (ii) average nucleotide identity (ANI) calculation via JSpeciesWS web service (http://jspecies.ribohost.com/jspeciesws/).

Determination of the Cr(VI) reducing capacity of strain M28T

Cr(VI) tolerance and reduction test of strain M28T

Strain M28T cells in the logarithmic growth phase were inoculated into LB liquid medium supplemented with varying initial Cr(VI) concentrations (25–150 mg/L) and incubated at 30 °C. Samples were collected periodically, with subsequent quantification of residual Cr(VI) in the supernatant using the diphenylcarbazide colorimetric method (GB 7467-87, 1987). Biomass monitoring was conducted via optical density measurements at 600 nm (OD600) using a UV-Vis spectrophotometer (T6 New Century) following resuspension of centrifuged cell pellets.

Effect of different T, pH value and electron donor on Cr(VI) reduction

Cells in the logarithmic growth phase were inoculated into LB liquid medium containing 50 mg/L Cr(VI). Experimental conditions included: Temperature gradients (28 °C, 30 °C, 35 °C, 37 °C, 40 °C), pH gradients (6.0–8.5 in 0.5 increments), Electron donor types (lactose, glucose, sucrose, fructose, glycerin, sodium pyruvate, Sodium lactate, and sodium acetate) at 5.0 g/L, Electron donor concentrations (0–10 g/L in specified gradients). Samples were collected at 6-h intervals to measure residual Cr(VI) concentration and the biomass of strain M28T.

Comparison of Cr(VI) reduction ability of strain M28T before and after optimization

Strain M28T was cultivated under both original and optimized growth conditions. Samples were collected and monitored at 6-h interval during the cultivation period. Biomass accumulation was quantified via OD600 measurements, while chromium detoxification capacity was assessed through diphenylcarbazide-based Cr(VI) spectrophotometric determination.

Results and the discussion

Several microbial strains exhibiting tolerance to Cr(VI) were successfully isolated from soil of decayed wheat straw through systematic screening. Phylogenetic analysis based on 16S rRNA gene sequences demonstrated that strain M28T showed less than 98.62% similarity to all validated type strains, suggesting its potential as a novel species. To establish its taxonomic position, a polyphasic taxonomic approach was employed to further determine its taxonomic status, including phylogenetic analysis, phenotypic characteristics, physiological and biochemical characteristics and chemical composition analysis. Furthermore, the Cr(VI)-reducing properties of the strain was also explored.

Identification of strain M28T

The 16S rRNA gene sequence of strain M28T was determined through PCR amplification followed by Sanger sequencing (1,386 bp), with additional verification through whole-genome sequencing-derived 16S rRNA sequence extraction (1,523 bp). Comparative analysis confirmed complete sequence identity between these two methodological approaches. The validated 16S rRNA sequence has been deposited in GenBank under accession number ON923955. According to the results of NCBI and EzBioCloud alignments, the type strains with the highest similarity to strain M28T were Microbacterium thalassium JCM 12079T (98.61%), M. resistens NBRC 103078T (98.48%), M. pullorum Sa4CUA7T (98.41%), M. testaceum NBRC 12675T (98.41%), M. keratanolyticum IFO 13309T (98.41%), M. allomyrinae NBRC 115127T (98.27%), M. aquimaris JCM 15625T (98.25%), and M. terricola JCM 14903T (98.25%). According to the current general standards of prokaryotic taxonomy (Kim et al., 2014), 16S rRNA gene sequence similarity below 98.65% is generally considered one of the important molecular indicators for distinguishing different species. In this study, the similarity between strain M28T and its closest relative was 98.61%, which is below the recommended threshold. This preliminary finding suggested that M28T may represent a new species.

Analysis of phylogenetic tree constructed based on 16S rRNA gene sequences (Fig. S1) showed that strain M28T clustered with three Microbacterium species (M. aquimaris JCM 15625T, M. resistens NBRC103078T, M. testaceum NBRC12675T). However, strain M28T formed a distinct phylogenetic lineage with significant branch separation from these reference species. Based on these results, it is clear that strain M28T is a member of the genus Microbacterium and might be a new species.

Phylogenetic trees were constructed using core genes and whole-genome sequences to clarify the taxonomic position of strain M28T (Fig. 1). In all analyses, strain M28T formed a stable cluster with M. profundi Shh49T and M. murale CCM7640T, yet maintained an independent phylogenetic branch distinct from other members of the genus Microbacterium. These phylogenomic results corroborated the 16S rRNA gene sequence similarity and topology of the 16S-based phylogenetic tree, collectively confirming strain M28T as a novel member of the genus Microbacterium.

Figure 1 Phylogenetic trees based on whole-genome and core genes.

(A) Whole-genome-based phylogeny constructed using TYGS, showing the phylogenetic position of strain M28T and its relatives; (B) neighbor-joining phylogenetic tree.

To definitively establish the taxonomic position of strain M28T, whole-genome-based average nucleotide identity (ANI) and digital DNA-DNA hybridization (dDDH) analyses were performed against its closest phylogenetic neighbors within the genus Microbacterium. The calculated OrthoANIu (less than 78.92%) and dDDH (less than 22%) values (Table 1) were significantly below the established species delineation thresholds (ANI ≥95–96%; dDDH ≥70% as per the code of nomenclature of prokaryotes). These genomic metrics conclusively demonstrate that strain M28T represents a novel species within the genus Microbacterium.

Table 1 ANI and dDDH values calculated by genomic sequence of strain M28T and related type strains of Microbacterium.

Alignment of strain M28T	OrthoANIu values with M28T (%)	dDDH values with M28 (%) formula 2	16S rRNA similarity (%)	Difference in % G+C	
M. testaceum NBRC 12675T	74.89	20.20%	98.32	0.74	
M. Algeriense G1 DSM 109018T	78.85	21.90%	98.18	0.8	
M. pleivorans NBRC 103075T	74.47	19.70%	98.18	0.22	
M. paraoxydans DSM 15019T	78.88	21.80%	98.11	0.26	
M. profundi Shh49T	78.92	22.00%	98.18	2.22	
M. resistens NBRC 103078T	77.66	21.30%	98.40	2.41	
M. murale CCM 7640T	78.78	21.8%	--	2.23	
Note:

The following is a list of strains with GenBank accession numbers for genome sequences in parentheses: 1. M. testaceum NBRC 12675T (GCA_006539145.1); 2. M. algeriense G1 DSM 109018T (GCA_008868005.1); 3. M. oleivorans NBRC 103075T (GCA_001552475.1); 4. M. paraoxydans DSM 15019T (GCA_900105335.1); 5. M. profundi Shh49T (GCA_000763375.1); 6. M. resistens NBRC 103078T (GCA_001552355.1); 7. M. murale CCM 7640T (GCA_014635185.1).

Strain M28T forms small, pale yellow colonies on LB agar plates and exhibits optimal growth at 25–30 °C and pH 7.0, with tolerance to 8% NaCl. It is strictly aerobic and shows positive results for catalase, urease, gelatin liquefaction, and nitrate reduction, but negative for methyl red and Voges–Proskauer tests. The strain metabolizes specific carbon sources such as dextran, maltose, trehalose, and gentiobiose. The Biolog similarity between strain M28T and M. aquimaris JCM 15625T was 70.9%, far below the 95% species delineation threshold, confirming significant phenotypic divergence. (Detailed phenotypic, physiological and biochemical characteristics, as well as chemical composition analyses of strain M28T can be found in the Supplemental Materials).

Polyphasic taxonomic evaluation (phylogenetic, phenotypic, physiological, biochemical, and chemical characterization) showed that strain M28T had both similarities and significant differences with the related type strains of genus Microbacterium. The polyphasic taxonomic evidence conclusively establishes strain M28T as a novel species within the genus Microbacterium.

Functional genome analysis

The whole-genome sequence of strain M28T was determined and analyzed on the cloud platform (https://cloud.majorbio.com/page/tools.html). The complete genome of strain M28T comprises a circular chromosome of 3.37 Mb with a G+C content of 68.76% (Fig. 2), deposited in GenBank under accession number CP107546. Genomic annotation revealed 3,215 protein-coding sequences and 52 RNA-coding genes, including 46 tRNAs, and two copies each of 5S, 16S, and 23S rRNA genes organized in ribosomal operons.

Figure 2 Circle genome mapping of strain M28T.

Gene Ontology (GO) annotation of strain M28T classified its functional genes into three primary domains: biological processes (BP), cellular components (CC), and molecular functions (MF). Comparative analysis against the GO database identified 812 genes annotated to BP, with predominant roles in transcriptional regulation/DNA-templated processes (75 genes), transmembrane transport (75 genes), and translation (56 genes). Among the 765 genes assigned to CC, the majority were associated with membrane components (493 genes), cytoplasmic localization (152 genes), and plasma membrane-associated functions (149 genes). For MF, 1,366 genes were functionally annotated, predominantly encoding ATP-binding (199 genes), DNA-binding (180 genes), metal ion-binding (100 genes), and hydrolase activity (90 genes). These annotations reflect critical functional modules supporting cellular growth and metabolic processes. Notably, the significant enrichment of metal ion-binding genes suggests a potential mechanistic link to this strain’s resistance to metal ions, aligning with its observed phenotypic traits.

Members of the genus Microbacterium exhibit broad environmental adaptability, thriving in diverse habitats including extreme niches, which reflects their genetic diversity and metabolic versatility. Functional annotation of strain M28T’ s genome identified a comprehensive suite of stress response systems, encompassing resistance mechanisms against metal ions (including zinc/manganese transporters, copper/nickel/cobalt resistance proteins, arsenic detoxification modules, and mercury reductase complexes; Table S1); diverse antibiotic resistance determinants (such as bleomycin/puromycin-specific resistance proteins, the multidrug efflux protein Stp (gene 0540), the fosfomycin resistance protein AbaF (gene 2285), and a chloramphenicol resistance protein (gene 2315)); alongside key enzymes for oxidative stress defense (including catalase, superoxide dismutase, and thioredoxin reductase).

Notably, within the metal resistance annotations, genes involved in c-type cytochrome biogenesis were identified (ResB gene 2521, CcdA gene 2522, and the qcrABC operon genes 1960–1962). Based on studies of the acidophilic bacterium Acidiphilium cryptum JF-5 demonstrating the involvement of its c-type cytochrome in Cr(VI) reduction (Magnuson et al., 2010), the Cr(VI) reduction capability of strain M28T may be associated with this finding. Furthermore, Cr(VI) reduction capacity has been confirmed in other members of the genus Microbacterium, exemplified by strain Microbacterium Cr-07 (Liu et al., 2012), which utilizes a glutathione-mediated non-enzymatic reduction mechanism—a pathway distinct from the one potentially employed by M28T. Crucially, genomic analysis of strain M28T detected no homologous sequences for the extensively studied Cr(VI) reductase genes reported in the literature (such as the ChrR family (Shi et al., 2023), YieF family, NfsA nitroreductase (Ackerley et al., 2004), or thioredoxin reductase). This absence strongly suggests that Cr(VI) tolerance and reduction in strain M28T likely depend on novel or atypical mechanisms.

Potential alternatives include unannotated genes not yet associated with chromium reduction; functional redundancy conferred by other reductases; or a predominant reliance on non-enzymatic reduction pathways, potentially analogous to the glutathione- or metabolite-mediated mechanism observed in strain Cr-07. Based on protein functional information derived from gene annotation, we speculate that cysteine-rich domains in surface proteins, which are structurally analogous to metallothioneins, may chelate extracellular Cr(III) through their thiol groups, thereby limiting the entry of toxic chromium ions (Wu et al., 2024). Meanwhile, mercury reductase homologous genes containing a conserved NADPH-binding domain could facilitate the reduction of Cr(VI) to Cr(III), while antioxidant enzymes such as superoxide dismutase and catalase may help mitigate chromium-induced oxidative stress (Hu et al., 2021). Together, these systems are hypothesized to function synergistically within an integrated ‘extracellular chelation-transmembrane reduction-intracellular detoxification’ pathway. This coordinated mechanism offers a compelling explanation for the efficient Cr(VI) removal observed even in the absence of classical chromate reductases. Furthermore, highly efficient tolerance mechanisms (e.g., efflux, chelation, or cell membrane fortification) may mitigate intracellular chromium stress, thereby enabling these non-canonical or less efficient reduction pathways to function effectively.

Beyond these stress adaptation capabilities, genomic annotation further elucidated the genetic basis for organic pollutant degradation in strain M28T. This includes 10 genes annotated within pathways for aromatic compound degradation (specifically benzoate, catechol, and phenylacetate). Concurrently, carbohydrate-active enzyme (CAZy) annotation results (Table S2) confirmed the presence of a diverse repertoire of lignocellulose-degrading enzymes. These encompass key enzymes for hemicellulose degradation (e.g., acetylxylan esterase, endo-1,4-β-xylanase, β-xylosidase, β-mannosidase, endo-1,3-β-xylanase), cellulose degradation (e.g., cellobiose dehydrogenase, endo-1,4-β-glucanase), and lignin degradation (e.g., vanillyl-alcohol oxidase, manganese peroxidase, versatile peroxidase, lignin peroxidase, and peroxidase).

In summary, the strain M28T genome harbors a rich genetic repertoire conferring resistance to diverse environmental stresses (including heavy metals, antibiotics, and oxidative stress) and the capacity to degrade complex organic pollutants (such as aromatic compounds and lignocellulose). These combined attributes underpin its unique and highly adaptive physiological profile.

Cr(VI) reduction characterization of strain M28T

The Cr(VI) reduction characteristics of strain M28T were evaluated across a concentration gradient (25–150 mg/L), and the effects of different initial Cr(VI) concentrations on the biomass and Cr(VI) reduction rate of strain M28T are shown in Fig. 3. Figure 3A demonstrates an inverse correlation between Cr(VI) concentration and bacterial growth, complete growth inhibition occurred at 150 mg/L, establishing 125 mg/L as the maximum tolerable concentration. It can be seen from Fig. 3B that the reduction rate of Cr(VI) gradually decreased with the stepwise increase of Cr(VI) concentration. When the initial concentration of Cr(VI) was 25 mg/L, the reduction rate of Cr(VI) is as high as 85.90%. When the Cr(VI) concentration was increased to 50, 75, 100 and 125 mg/L, the reduction rate was decreased to 58.29%, 40.33%, 29.86% and 28.32% respectively. This synchronized decrease in both biomass and metabolic activity indicates that chromium toxicity primarily manifests through growth suppression coupled with functional impairment.

Figure 3 Growth of strain M28T for different initial Cr(VI) contents (A) and reduction rate of Cr(VI) (B).

The effect of different temperature on biomass accumulation and Cr(VI) reduction capacity of strain M28T was presented in Fig. 4. When the initial concentration of Cr(VI) was 50 mg/L, the growth of strain M28T was affected by both high and low temperatures, with the maximum growth at 37 °C (Fig. 4A). Cr(VI) reduction rate revealed a temperature optimum at 37 °C (67.67% efficiency), with progressive efficiency declines observed at suboptimal temperatures: 43.6% (28 °C), 50.33% (30 °C), 64.33% (35 °C), and 41.33% (40 °C) after 84 h incubation (Fig. 4B). It was found that 37 °C is the most suitable temperature for the growth and the reduction of Cr(VI) for strain M28T, the enzymatic activity and metabolic pathways of microbial strains are susceptible to inhibitory effects when exposed to either excessively high or low temperatures, primarily due to protein denaturation and conformational alterations.

Figure 4 The effects of different culture temperature (A/B) and pH values (C/D) on growth and Cr(VI) reduction efficiency of strain M28T.

The effects of different pH value on growth and Cr(VI) reduction capacity of strain M28T were shown in Fig. 4. Figures 4C and 4D demonstrated optimal microbial performance at pH 7.0, with peak biomass accumulation and Cr(VI) reduction efficiency (84.84%), whereas both acidic (pH 6.0–6.5) and alkaline (pH 7.5–8.5) conditions resulted in significantly decreased metabolic activity. The reduction efficiency systematically declined to 67.00% (pH 6.0), 75.67% (pH 6.5), 49.51% (pH 7.5), 40.33% (pH 8.0), and 37.33% (pH 8.5), exhibiting 9–47% performance loss compared to the neutral optimum. Cr(VI) reduction is fundamentally an enzymatically catalyzed process, where pH fluctuations critically regulate biocatalytic efficiency through perturbations in active-site protonation states and enzyme tertiary structural stability. These interdependent physicochemical alterations directly modulate electron transfer kinetics and substrate binding affinity, ultimately governing Cr(VI) detoxification performance.

The effect of different electron donors on the biomass and Cr(VI) reduction rate of strain M28T were shown in Fig. 5. Biomass production demonstrated marked electron donor dependence, the growth of strain M28T supplemented with fructose and glycerol were higher than other substrates (Fig. 5A). The Cr(VI) reduction profile revealed distinct performance variations, where sucrose and glycerol achieved peak efficiency (97.67–98.00%), closely followed by lactose, sodium pyruvate, and fructose (96.67% each). In contrast, glucose and sodium lactate showed intermediate reduction rates (69.67–82.33%), while the electron donor-free control exhibited only 40.33% removal efficiency (Fig. 5B). Notably, temporal analysis showed glycerol and sucrose achieved 90% reduction within 24 h, whereas lactose, sodium pyruvate and fructose required 48 h to reach comparable levels. This electron donor preference aligns with but differs from prior observations-Pseudochrobactrum saccharolyticum W1 exhibited peak activity with 4.0 g/L sodium lactate (Li et al., 2019), while Bacillus sphaericus AND 303 showed glucose preference (Pal, Dutta & Paul, 2005).

Figure 5 The effects of different electron donors (A/B) and different concentrations of sucrose (C/D) on growth of strain M28T and Cr(VI) reduction rate.

The Cr(VI) reduction efficiency generally increases with electron donor supplementation until reaching a plateau phase, primarily constrained by the cellular reductase capacity. This saturation occurs when electron donor concentrations exceed the binding potential of enzymatic active sites, rendering additional donors ineffective for further enhancement. The above experimental result identified sucrose as the optimal electron donor for enhancing both biomass accumulation and Cr(VI) reduction in strain M28T. As demonstrated in Fig. 5C, biomass production exhibited a dose-dependent response to sucrose supplementation, with cell density progressively increasing across the tested concentration gradient. Concurrently, Cr(VI) reduction efficiency displayed threshold kinetics (Fig. 5D), where sucrose concentrations ≥2.5 g/L enabled near-complete pollutant removal (96.01–99.22%) within 30 h. In contrast, suboptimal concentrations of 1 g/L achieved only 74.25% reduction, while the electron donor-free control showed minimal activity (26.00%). It can be seen that 2.5 g/L sucrose supplemention was the most suitable electron donor concentration, simultaneously maximizing microbial growth and enzymatic detoxification capacity.

Process optimization induced significant physiological enhancements (Fig. 6). Under refined conditions (50 mg/L Cr(VI) concentration, 37 °C, pH 7.0, 2.5 g/L sucrose supplemented), strain M28T exhibited 2.6-fold greater biomass accumulation and achieved 99.00% Cr(VI) reduction within 24 h—a 7.07-fold increase over unoptimized performance (14.00%). The 85% efficiency differential between optimized and control groups confirms the critical role of parameter optimization in microbial metal detoxification systems.

Figure 6 Growth (A) and Cr(VI) reduction (B) of strain M28T for pre-optimized and post-optimized conditions.

Conclusion

This study reports the isolation and characterization of Microbacterium triticisoli sp. nov. M28T (CCTCC AA 2022021T = JCM 35796T), a novel chromium-reducing bacterium isolated from chromium-contaminated soils in Gansu Province, China. Polyphasic taxonomic analysis confirmed its novel species status within the Microbacterium genus. The isolate demonstrated exceptional Cr(VI) detoxification capabilities, achieving complete reduction (>99% efficiency) of 50 mg/L Cr(VI) within 24 h under optimal conditions (37 °C, pH 7.0, 2.5 g/L sucrose supplemented). Remarkably, it maintained metabolic activity and growth at 125 mg/L Cr(VI). Genomic analysis revealed multiple metal resistance systems, antibiotic resistance determinants and oxidative stress defense genes, explaining its exceptional environmental adaptability. These findings establish strain M28T as a multifunctional candidate for in situ bioaugmentation, providing both mechanistic insights into microbial chromium resistance and practical solutions for composite heavy metal pollution remediation.

Taxonomic conclusions

Based on phylogenetic, phenotypic and genomic characterisation, strain M28T has both commonalities and differences with related type strains of the genus Microbacterium. These features support the classification of strain M28T as a new species of the genus Microbacterium, the name Microbacterium triticisoli sp. nov. is proposed.

Description of Microbacterium triticisoli sp. nov.

Description of Microbacterium triticisoli sp. nov. Microbacterium triticisoli (tri.ti.ci.so’li. L. neut. n. triticum, wheat; L. neut. n. solum, soil; N.L. gen. n. triticisoli, of wheat soil).

Cells are Gram-stain-positive, aerobic, non-motile, catalase-positive and oxidase-negative. The cells were short rods (0.5–1.0 × 0.8–2.0 μm), colonies on LB agar after 3 days incubation at 30 °C are circular, convex with regular edges and light yellow. Growth occurs at 15–38 °C (optimum 30 °C), pH 5.0–8.5 (optimum pH 7.0), and in the presence of 0–8.0% (w/v) NaCl (optimum growth at 1.0% NaCl).

In biochemical characterization, strain M28T showed negative reactions for starch hydrolysis, Tween 40 degradation, milk coagulation, cellulose hydrolysis, peptonization, methyl red (MR), and Voges-Proskauer (V-P) tests. Enzyme activities (API ZYM) revealed positive reactions for: alkaline phosphatase, esterase (C4), leucine arylamidase, acid phosphatase, naphthol-AS-BI-phosphohydrolase, β-galactosidase, α-glucosidase, β-glucosidase, N-acetyl-β-glucosaminidase and α-mannosidase; weakly positive for esterase lipase (C8), valine arylamidase and α-galactosidase; negative for lipase (C14), cystine arylamidase, trypsin, α-chymotrypsin, β-glucuronidase and α-fucosidase.

Carbon source utilization (GEN III MicroPlate) showed positive assimilation of: dextrin, D-maltose, D-trehalose, D-cellobiose, gentiobiose, sucrose, D-turanose, stachyose, D-raffinose, D-melibiose, α-D-glucose, D-mannose, D-fructose, D-galactose, L-rhamnose, L-glutamic acid, pectin, p-hydroxy-phenylacetic acid, D-mannitol and acetic acid; weak utilization of: α-D-lactose, D-salicin, N-acetyl-D-glucosamine, N-acetyl-D-galactosamine, inosine, glycerol, glycyl-L-proline, L-aspartic acid, D-gluconic acid, L-lactic acid, α-hydroxy-butyric acid, α-keto-butyric acid and propionic acid. Additional carbohydrate tests (API 50CH) confirmed positive reactions for 29 substrates including glycerol, D-arabinose, and D-glucose.

In the API 50CH system, strain M28T is found to be positive for glycerol, D-arabinose, L-arabinose, D-ribose, D-xylose, methyl-β-D-xylopyranoside, D-galactose, D-glucose, D-fructose, D-mannose, arbutin (weak), L-rhamnose, D-lactose, D-mannitol, methyl-α-D-mannopyranoside, methyl-α-D-glucopyranoside, amygdalin (weak), esculin, salicin(weak), D-cellobiose, D-maltose, D-melibiose (weak), D-saccharose, D-trehalose, D-melezitose, D-raffinose (weak), starch (weak), glycogen (weak), gentiobiose, D-turanose, L-fucose.

Strain M28T was found to have iso-C15:0 (11.67%), anteiso-C15:0 (35.60%), iso-C16:0 (28.44%), and anteiso-C17:0 (14.77%) as the major fatty acids. Polar lipids comprised phosphatidylglycerol (PG), diphosphatidylglycerol (DPG) and an unknown glycolipid (GL). The major menaquinones were MK-10, MK-11 and MK-12. The whole genome size of strain M28T is 3.37 Mb with 68.76% of DNA G+C content.

The type strain, M28T (CCTCC AA 2022021T=JCM 35796T), isolated from surface soil obtained from decayed wheat straw in Qingyang City, Gansu Province, China. The 16S rRNA gene and whole genome sequences are deposited in GenBank under accession numbers ON923955 and CP107546, respectively.

Supplemental Information

Supplemental Information 1 Methods and results for p henotypic, physiological and biochemical characteristics, and chemical component of strain M28 T.

Supplemental Information 2 Neighbor-joining phylogenetic tree based on 16S rRNA gene sequences, showing the phylogenetic position of strain M28T among related strains.

Supplemental Information 3 Heavy metal related genes in strain M28T.

Supplemental Information 4 Genes related to lignocellulose degradation annotated in M28T.

Supplemental Information 5 Culture collection certification-CCTCC.

Supplemental Information 6 Strain Collection Certification-JCM.

Supplemental Information 7 Raw Data (Figures 3-6).

The authors are deeply indebted to Professor Zuo Yi for conducting fatty acid measurements.

Additional Information and Declarations

Competing Interests

The authors declare that they have no competing interests.

Author Contributions

Yu Qing conceived and designed the experiments, performed the experiments, analyzed the data, prepared figures and/or tables, and approved the final draft.

Jiewei Tian analyzed the data, authored or reviewed drafts of the article, and approved the final draft.

Zhenhua Ma conceived and designed the experiments, performed the experiments, analyzed the data, prepared figures and/or tables, and approved the final draft.

Miao Tang performed the experiments, analyzed the data, prepared figures and/or tables, and approved the final draft.

Xiufeng Long conceived and designed the experiments, analyzed the data, prepared figures and/or tables, authored or reviewed drafts of the article, and approved the final draft.

Data Availability

The following information was supplied regarding data availability:

The 16S rRNA gene and whole genome sequences are available at GenBank: ON923955 and CP107546.

New Species Registration

The following information was supplied regarding the registration of a newly described species:

Microbacterium triticisoli sp. nov. M28T (=CCTCC AA 2022021T =JCM 35796T)

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
