# Peer review of "Taxonomic identification, genomic analysis, and optimized chromium(VI) bioreduction by Microbacterium triticisoli sp. nov. M28T"

_PeerJ, doi:10.7717/peerj.20192_

## Round 0.1 · original submission · Major Revisions

Please revise the manuscript by following the reviewers' comments.

Reviewer 1 ·

Basic reporting

COMMENTS FOR THE AUTHOR’S CONSIDERATION:

This was an interesting piece of work. The authors appeared to have enthusiasm for new technology. The paper could be shortened. The title could be changed. The paper seemed to contain two projects. The chromium remediation was the key subject in the work and not taxonomy. It could be appropriate to present the significance of the strain first and the strain identification/characterization second. (The isolate may be similar to a species of Microbacterium.) Some of the tables and figures should be eliminated.

Line 35. Should anthropogenic be explained?

Line 46. The text may be awkward here.

Line 73. Do emendations need to be explained?

Line 81. The language may be awkward in this area.

Line 84. Is there a repeat?

Line 104. Watch spacing.

Line 108. Expound on agar concentration of 20 g. Fifteen or 16 g is common.

Line 114. Perhaps briefly explain standardized PCR conditions. How was the DNA
extracted?

Line 137. Should there be two sentences here?

Line 201. Would it be appropriate to elaborate on the appearances of the colony types
initially isolated from the soil?

Line 182. Would it be appropriate to expound a little on the method used to measure the
Cr(VI)?

Line 183. Was biomass and optical density the same thing in this paper? What kind of
spectrophotometer or colorimeter was used?

Line 210. Would it be appropriate to provide information on percent DNA sequence
relatedness needed for designation of a new species?

Line 252. Provide the anaerobic technique used.

Line 256. Simplify the list. Do not repeat the same data in the text and in a table or
Figure. What percent matching of species did the Biolog system produce?

Line 263. Elaborate on rationale of enzyme screening. Do not repeat the same data in
the text and in a table or figure.

Line 268. Elaborate on rationale of API assay. Do not repeat the same data in the text
and in a table or figure.

Line 274 Elaborate on rationale of chemical assay. Do not repeat the same data in the
text and in a table or figure.

Line 287. Part of the genome analysis could be removed. Genetic Information that is
related to chromium remediation might be suitable.

Line 350. How was reduction measured? Was microbial growth representing reduction?

Check tables and figures. The information in in Table 2 may be crowded at the bottom.

Was the use of a control reference organism (perhaps Bacillus or Pseudomonas) considered?

Experimental design

Satisfactory (see above)

Validity of the findings

Satisfactory (see above)

Reviewer 2 ·

Basic reporting

The authors provided multiple evidences (eg sequence identity, physiology, cell structure, and biochemical property, and genome characteristic) that Strain M28 T isolated in this study is a new species of Microbacterium. I commend the authors for such amount of work they have done here. I would recommend publication of this study after addressing some minor edits.

Minor edits. For introduction, there were some citations (eg Lines 43 Hossini, Lines 45 Gerogaki, etc) that were seem to be misplaced. please review the citations and place after the sentences respectively throughout the manuscript.

Experimental design

In lines 113-119, Provide the primer sequences details, or include in supplementary information. I would also suggest to include the full PCR conditions and type of sequencing, platform used in the main method section.

In method sections 2.2, some important information were lacking
lines 124- washed sequentially with what?
lines 125- ethanol gradient, specify the concentrations used.
Lines 127, specify the kit.

For lines 132-136, include a supplementary information on the methods for assays employed in this study. A written method or protocol is important for other researcher as reference.

"Voges-Proskauer (acetoin detection) and methyl red (acid production) assays, catalase activity, oxidase activity, and substrate utilization profiles (including cellulose hydrolysis, gelatin liquefaction, starch degradation, Tween hydrolysis, milk coagulation and peptonization, urea hydrolysis, hydrogen sulfide production, and nitrate reduction) were assessed according to theliterature (Gonzalez et al., 1978)"

Lines 146, what is TLC?

Lines 150, include the specific details of kits used for DNA extraction, and a brief protocol how it was done. how many cells were extracted etc and which Illumina platform was used.

For lines 153-163, include details how many bases were trimmed before doing assembly, how many libraries were assembled.

Include citations for all these databases Gene Ontology (GO), Non-Redundant Protein (NR), Swiss-Prot, Pfam, Clusters of Orthologous Groups (COG), and Kyoto Encyclopedia of Genes and Genomes (KEGG) databases.
Lines 164-165, inclde the accession number here.

Validity of the findings

In the discussion part, it is important to highlight and discuss if you see some putative genes that could explain the biolchemical proterties you observed, as well as genes that have putative functions for chromium bioremediation. the later should be discussed and highloghted.

In lines 324-327 "M28T detected no homologous
325 sequences for the extensively studied Cr(VI) reductase genes reported in the literature (such as the 326 ChrR family (Shi et al.,2023), YieF family, NfsA nitroreductase (Ackerley et al.,2004), or 327 thioredoxin reductase). This absence strongly suggests that Cr(VI) tolerance and reduction". are there any other genes in the genome that have similar functions?

---

## Round 0.2 · Minor Revisions

Please revise the manuscript by following the reviewer's comments.

Reviewer 1 ·

Basic reporting

COMMENTS FOR THE AUTHOR’S CONSIDERATION (SECOND REVIEW):

The reduction of chromium by bacteria is an interesting and timely subject. The authors performed a great deal of work and appeared to be especially interested in the molecular genetics of bacteria. Some small improvements were made in this draft of the paper, but the document remained basically the same.

The paper may need to be shortened. There may be enough information in the paper
for a reader to lose track. In a typical paper much of the data that was made in the research lab never makes it to publication. There may be sections of the paper that may not be directly associated with a purpose. (Some of the text is just information.)

Consider making two papers. One paper could be on the reduction of chromium. And the other paper could be on the genetics and nomenclature of the microorganism.
It might be appropriate to postpone naming the microorganism. One definition of a species is a group of strains that have characteristics that make them different from other groups of strains. Have any other strains of this novel chromium-reducing microorganism been isolated by other investigators? Clinical labs have reported that some isolates of bacteria appear to be between two known organisms, but they were not designated as a new species. With some other bacteria, the name changing of an organism has gone on for decades!

Was biomass the correct word to use in the text? What was measured in experiments? What was the definition of biomass?

Experimental design

Satisfactory (see above)

Validity of the findings

Satisfactory (see above)

Reviewer 2 ·

Basic reporting

The authors have already addressed all my comments. I strongly recommend the publication of such an extensive work.

Experimental design

none

Validity of the findings

none

Additional comments

none

---

## Round 0.3 · accepted · Accept

Based on the revised version of the manuscript and the reviewer's comments, I endorse the publication of the manuscript.